# Distribution and Habitat Preferences of the Newly Rediscovered *Telmatogeton magellanicus* (Jacobs, 1900) (Diptera: Chironomidae) on Navarino Island, Chile

**DOI:** 10.3390/insects11070442

**Published:** 2020-07-14

**Authors:** Felipe Lorenz Simões, Tamara Contador-Mejías, Javier Rendoll-Cárcamo, Carolina Pérez-Troncoso, Scott A. L. Hayward, Edgar Turner, Peter Convey

**Affiliations:** 1British Antarctic Survey, Cambridge CB3 0ET, UK; 2Department of Zoology, University Museum of Cambridge, Cambridge CB2 1TN, UK; ect23@cam.ac.uk; 3Sub-Antarctic Biocultural Conservation Program, Wankara Sub-Antarctic and Antarctic Freshwater Ecosystems Laboratory, Universidad de Magallanes, Puerto Williams 6350000, Chile; contador.tamara@gmail.com (T.C.-M.); javier.rendoll@gmail.com (J.R.-C.); ca.pereztroncoso@gmail.com (C.P.-T.); 4Millennium Nucleus of Invasive Salmonids, INVASAL, Concepción 4030000, Chile; 5School of Biosciences, University of Birmingham, Birmingham B15 2TT, UK; s.a.hayward@bham.ac.uk

**Keywords:** algae, brachypterous, flightless, intertidal, life history, Magellanic, non-biting midge, polar, sub-Antarctic

## Abstract

The habitat of the intertidal flightless midge *Telmatogeton magellanicus* (Jacobs, 1900) is described for the first time from the northern coast of Navarino Island, Tierra del Fuego, Chile. Additionally, we report the first observations of adult behaviour in the wild. We delineate the species’ distribution across three tidal zones (high, mid and low), and identify substrate characteristics that favour the presence of the midge. The mid-tide zone was the key habitat utilized by *T. magellanicus*, with lower densities in the low-tide zone and no presence in the high-tide zone. There was a strong association between the presence of larvae and filamentous algae, especially *Bostrychia* spp. and, to a lesser extent, *Ulva* spp., as well as between larvae and the presence of larger, more stable boulders. As a result, the species’ overall distribution was widespread but patchy. We suggest that the main limiting factor is the relative humidity experienced in different habitats. One of the most striking features of the behavioural observations during data collection was the extremely active adults, which suggests high energy expenditure over a very short period of time. This may be due to the limited time available to find mates in a single low-tide period, when adults have about three hours after emerging from the pupa to complete mating and oviposition before inundation by the tide. The data presented here provide a baseline for future studies on this species’ ecology, phenology, physiology and general biology.

## 1. Introduction

### 1.1. Terrestrial Invertebrates in the Magellan Strait

The Magellanic region of Tierra del Fuego is not strictly considered to be part of the sub-Antarctic [1,2], which is defined by the almost complete absence of true terrestrial vertebrates and woody plants. However, because it shares many climatic and environmental features with this region, it is often referred to as the Magellanic sub-Antarctic [3,4,5]. The Magellanic region is still largely unexplored, with recorded biodiversity increasing mainly through the discovery of cryptic species and the activity of new research projects working in little-studied areas [2]. A major component of the plant life is endemic to the region [3,4,5], which is likely to have specific suites of associated invertebrates. There are no published syntheses of the invertebrate fauna of the region, either aquatic or terrestrial, although several authors have described endemic species of the latter: Coleoptera [6,7,8,9,10,11,12], Hymenoptera [13], Lepidoptera [14] and Neuroptera [15].

Discovering and characterising invertebrate diversity, distribution and habitat preference is an extremely important part of understanding the ecology of high latitude southern ecosystems. Invertebrates living in these habitats often display adaptations that allow them to tolerate extreme conditions. Therefore, understanding the physiology of resident invertebrate species provides the basis for comparisons across species or environments [16,17,18,19,20,21,22,23].

In the case of the Magellanic midge, *Telmatogeton magellanicus* (Jacobs, 1900), early taxonomic studies provided very little information about the species’ ecology or habitat requirements. Indeed, since the species’ original discovery and description, to our knowledge there have been no published studies addressing any aspect of its biology. Yet follow-up studies on *T. magellanicus* are extremely timely, as it forms part of a small group of key maritime and sub-Antarctic chironomid midge species that may help to clarify the historical biogeography of this region. This group includes *Belgica antarctica* Jacobs (1900), endemic to the Antarctic Peninsula and South Shetland Islands [24,25,26], and *Eretmoptera murphyi* Schaeffer (1914), originally described from and endemic to sub-Antarctic South Georgia and now introduced to the maritime Antarctic Signy Island [27,28,29,30,31]. Everatt et al. [32] concluded that the latter species was pre-adapted to harsher conditions than what currently prevail in its native South Georgia, where it is now known to be palaeoendemic [24,25] and, therefore, has the potential to invade the Antarctic Peninsula region [33].

### 1.2. Telmatogeton Magellanicus

The brachypterous midge *Telmatogeton magellanicus* (Figure 1) was originally classified by Jacobs [34], based on material collected on 19 December 1897, in the genus *Belgica*. This is a genus well-known for its type species *Belgica antarctica*, which is one of only two insect species currently native to parts of the Antarctic continent [2,35]. Rübsaamen [36] transferred the species to a new genus, *Jacobsiella*, based on examination of the original material, and expanded its description significantly. Edwards [37] synonymised *Jacobsiella* to *Halirytus* based on features of the tarsi and ovipositor, and raised the possibility of *H. magellanicus* and *H. amphibius* (Eaton, 1875) (a species from the sub-Antarctic Kerguelen archipelago in the Indian Ocean) being the same species; Edwards’ comparisons were based on examination of Eaton’s specimens of *H. amphibius* along with the description and images of *H. magellanicus*. Edwards [38] further addressed the taxonomic position of *Halirytus*, placing it in the “*Telmatogeton*” group and commenting that the former is in fact a reduced form of the latter. He considered the status of the two then-described species of *Halirytus* (with the later addition of *H. macquariensis* Brundin (1962)) and concluded that there was no evidence to doubt the validity of *H. magellanicus* as a separate species. There appears to have been no published reports of new collections of *H. magellanicus* since those originally examined by Jacobs [34]. Wirth [39] and Sublette and Wirth [40] further addressed taxonomy in the genus—with the latter considering *Halirytus* to be a brachypterous form of *Telmatogeton*, synonymising it to the latter. None of these taxonomic studies provided information about the habitat in which the midge was found, although it might be assumed that the species would require similar habitats or conditions to *T. amphibius*, *T. macquariensis* and other *Telmatogeton* species, which are typically found among filamentous algae attached to hard substrates in the intertidal and supralittoral zones [41,42,43,44].

Until recently, *T. magellanicus* was known only from the region of the Beagle Channel, and more specifically the type locality on Navarino Island, recorded as the “Grand Glacier Bay, Tierra del Fuego, English Channel, Chile” by Jacobs [34]. In 2015, the species was rediscovered in several intertidal locations across the northern coast of Navarino Island, Chile, and in the Cape Horn Islands; there is also a possibility of it being found along the coastline around Punta Arenas, Chile (S. Rosenfeld pers. comm.). New observations and collections in January 2020 confirmed the species to be present in the same intertidal habitats in the Argentinian Tierra del Fuego, in the locality of Puerto Almanza (T. Contador pers. obs.) and in Stanley Harbour in the Falkland Islands (P. Convey pers. obs.). However, no formal attempt has been made to characterise the species’ preferred habitat or its potential distribution, other than that it is found in the intertidal zone. Other intertidal insects living in similar habitats have been found to have behavioural and physiological rhythms linked to the tidal cycle through either endogenous clock-like mechanisms or direct external factors, which can range from environmental cues (such as temperature or sunlight) to the presence of key habitat features, such as algal food sources and substrate type (e.g., [45,46,47,48,49]).

In this study we document the distribution of *T. magellanicus* on the north coast of Navarino Island and describe aspects of the species’ behaviour in its natural habitat. We also categorise the local habitat and microenvironmental conditions where *T. magellanicus* is found. Through this survey we aim to better understand the biotic and abiotic drivers of *T. magellanicus* distribution across tidal zones.

## 2. Materials and Methods

### 2.1. Study Site Description

Fieldwork took place along the northern coast of Navarino Island (Figure 2), Tierra del Fuego, from 23 October to 28 November 2017, and consisted of single day visits to the selected bays (Figure 2), and several visits to Róbalo Bay (Figure 2 label ‘G’ and Figure 3). Additional monthly visits to the latter were made between late 2016 and early 2018. Air temperatures in this area range from as low as −12 °C in the winter to 26 °C during the summer (means of 6.0 °C across the year, 9.6 °C during the warmest month and 1.9 °C during the coldest [50]). The region is relatively dry, with the relative humidity (R.H.) averaging 69.3% but varying between 40.2% and 96.0%. Total precipitation for 2018 was 560.8 mm (a minimum of 23.2 mm in April and maximum of 89.9 mm in June). The climate is heavily influenced by prevailing westerly winds that can reach average speeds of up to 39 kts (in 2018). The average tidal range is 1.40 m, ranging from 0.15 m to 2.51 m [50,51]. At each visit, targeted searches were carried out to confirm the presence of adult *T. magellanicus*. We mapped the presence of *T. magellanicus* at each survey location in order to visualise the distribution of the species across Navarino Island. We also made notes of any aspects of adult behaviour that we observed on occasion.

### 2.2. Habitat Characterisation at Róbalo Bay

We selected Róbalo Bay to carry out finer-scale measurements of the environmental conditions influencing the abundance of *T. magellanicus*, owing to an abundant population being observed during initial field visits and to the site’s ease of access.

### 2.3. Environmental Variability within Low- and High-Tide Limits

To describe the habitat of *T. magellanicus* at Róbalo Bay, we divided the area into 28 transects (Figure 3). Transects were separated by 50 m from each other along the shore, and each ran in a straight line, perpendicular to the coast, from the low- to high-tide limits. At each transect, we randomly selected one area to deploy a 60 cm × 60 cm quadrat (sub-divided into 5 × 5 sub-quadrats—25 squares in total) at each of three tidal heights (low tide, mid tide and high tide) (Figure 4). The tide levels were determined by a combination of two factors: (i) recording the water level at high and low tide as predicted from average levels taken from tide tables [52]; and (ii) biological composition, as defined by Contador et al. [4]. Thus, we defined the high-tide level as the area between 2.5 m down to where molluscs (mainly bivalves) start to occur in abundance (around 1.5 m) and the mid tide from ~1.5 m to ~0.8 m, which is where the low-tide zone started with the presence of cirriped crustaceans and coralline algae. Upon starting the survey at each station, we measured air temperature, relative humidity (R.H.) and wind speed (1.5 m height) using a thermocouple logger (Hobo^®^ 4-channel UX120-014M) and anemometer (Kestrel 3000 Environmental Meter), respectively. We also recorded substrate surface temperatures and R.H. We then searched the 25 sub-quadrats for 1 min each, recording, through visual inspection and manipulation of the top layer of the substrate, the presence/absence of *T. magellanicus* larvae of any stage. We chose larvae for this study because they are less mobile than adults and their presence demonstrates that breeding is occurring, making them a more reliable indicator of true habitat requirements for the species.

Each quadrat was photographed for subsequent confirmation of the environmental components, namely, boulders (rocks larger than 26 cm in diameter), stones (rocks between 1 cm and 26 cm), gravel (crushed stone and any clustering of small stones up to 1 cm), sand, bivalves and water, as well as for the following marine algae: *Bostrychia* spp., *Ulva lactuca*, *U. intestinalis*, *Adenocystis* spp., *Porphyra* spp. and “other” (including Rhodophyta, *U. prolifera*, *Nothogenia* spp., *Macrocystis* spp., *Scytosiphon lomentaria* and large patches of mixed dead algae).

### 2.4. Statistical Analyses

We performed a Permutational MANOVA (PERMANOVA) with pair-wise tests to test for differences in the habitat composition of the 12 most frequent environmental variables between the three tide levels (low, medium and high) [53]. These variables, acquired through the quadrat methodology, were Boulder, Stones, Gravel, Sand, Bivalves, Water, *Bostrychia* spp., *Ulva lactuca*, *U. intestinalis*, *Adenocystis* spp., *Porphyra* spp. and Other (which includes other minor features, such as sporadic algal species or clumps of decomposing organic matter). The procedure was performed using normalized data to construct a resemblance matrix based on Euclidean distances with 9999 unique permutations, with “tide” as the only fixed factor included in the model. We calculated the contribution of each environmental variable at each tide level using the Similarity Percentages (SIMPER) procedure. The SIMPER procedure list was cut off when the accumulated contribution of each environmental variable reached 90%. Both analyses, PERMANOVA and SIMPER, were conducted using PRIMER-E v7 with the PERMANOVA+ add-on package [53,54].

We also compared the air and substrate temperatures, air and substrate R.H., as well as the average and maximum wind speeds between the three areas using Kruskal–Wallis tests. Post hoc paired Wilcoxon tests were used where significant differences were found. These tests and analyses were run with R (version 3.6.0) [55] in RStudio (Version 1.2.1335) [56].

### 2.5. Presence of T. magellanicus across the Three Tidal Zones

Proportional occurrence data in quadrats were tested for normality using Kolmogorov–Smirnov tests, but were non-normal, so we used a Kruskal–Wallis test, with post hoc paired Wilcoxon tests, to assess whether the occurrence of *T. magellanicus* differed between the tidal zones.

### 2.6. Prediction of T. magellanicus Presence within the Mid-Tide Zone

As the vast majority of *T. magellanicus* were found in the mid-tide zone, we carried out a more detailed analysis to assess how environmental factors influenced its presence in this zone. Here, we once again ran PRIMER-E v7 with the PERMANOVA+ add-on package [53,54], where abundances were transformed to presence/absence data and resemblance matrices created based on Euclidean distances [54]. Environmental data were normalized prior to generating the Euclidean resemblance matrices. Then, the combined influence of the environmental variables on the habitat preference of *T. magellanicus* was investigated using Distance-Based Linear Modelling (DistLM) in PERMANOVA+, using the “Best” selection criterion and adjusted R^2^ values. The output for the “Best” selection procedure in DistLM aims to provide the best 1-variable model, the best 2-variable model, and so on, on the basis of the chosen selection criteria [53]. DistLM seeks the most significant relationships between the similarity matrix and environmental variables by progressively modelling the matrix against the most influential variable, taking the residuals of that relationship, and then modelling the next most influential variable, and so on [54,57].

## 3. Results

### 3.1. Island-Wide Distribution

*Telmatogeton magellanicus* was present in most of the bays along the northern coast of Navarino Island (Figure 2). In some places, such as the “Casita” and “Fifth” Bays (Figure 2 labels ‘M’ and ’Q’, respectively), there were lower abundances of both adults and larvae. *T. magellanicus* shared the habitat with an unidentified winged species of *Telmatogeton* in Puente Guanaco Bay (Figure 2 label ‘I’), but this species was not recorded at any other location. Adults were recorded in almost every month with the exception of July and August, as we could not survey the area during this time. Adults were very active, moving rapidly while searching for mates. While mating was easily observed, we did not observe oviposition in the field, although females would readily oviposit in containers after capture (either on algae or on the surface of the container). Adults actively avoided water, using surface tension to move rapidly across the water surface only when this was unavoidable. Either as a consequence of entrapment in water or as result of post-mating death, the beach became littered with corpses of dead adults at the end of the tidal cycle that were subsequently washed into the sea as the tide rose back again.

### 3.2. Environmental Variability within Low- and High-Tide Limits

There was a fairly even distribution of variables across the tidal zones (Figure 5), but the PERMANOVA procedure showed that there were significant differences in habitat composition within the three tidal zones (mean squares = 62.237; pseudo-F = 5.7927; *p* = 0.0001). The post hoc tests indicated that significant differences existed between the high and low, the high and mid, and the low and mid tidal zones (Table 1).

The SIMPER procedure identified the % contribution of the 12 environmental variables that contributed 90% of the habitat composition across the three tidal zones (Table 2).

Differences in air temperature were significant between the three zones (χ^2^ = 10.814, d.f. = 2, *p* = 0.002), with temperatures in the mid tide being significantly higher than the high tide (*p* = 0.002). Air R.H. was not significantly different between zones (χ^2^ = 4.457, d.f. = 2, *p* = 0.108). Substrate surface R.H. was significantly different (χ^2^ = 16.386, d.f. = 2, *p* < 0.001), with R.H. in the high-tide zone being significantly lower than the low (*p* < 0.001) and mid zones (*p* = 0.020). Average wind speeds were significantly different between zones (χ^2^ = 9.301, d.f. = 2, *p* = 0.010), being higher in the high-tide zone than in the low zone (*p* = 0.011) (Figure 6).

### 3.3. Difference in the Presence of Telmatogeton magellanicus between the Three Tidal Zones

Larvae of *T. magellanicus* differed significantly in abundance between the three tidal zones (Kruskal–Wallis, χ^2^ = 23.138, d.f. = 2, *p* < 0.001), being more common in the mid-tide zone (90% of all the sub-quadrats had confirmed presence) than in the low-tide (9.67%) and high-tide (0.33%) zones (Figure 7).

### 3.4. Prediction of the Presence of T. magellanicus within the Mid-Tide Zone

Distance-Based Linear Modelling (DistLM) analysis of the impact of the 12 environmental variables on the habitat preference of *T. magellanicus* indicated that, considered separately, boulders, sand, bivalves, *Bostrychia* spp., *U. lactuca*, *U. intestinalis* and *Adenocystis* spp. were significantly positively associated with the presence of *T. magellanicus* (Table 3). However, when all combinations of variables were considered, the “Best” modelling procedure showed that the presence of a combination of boulders (5%), gravel (2%), sand (9%), bivalves (12%), *Adenocystis* spp. (6%), *Bostrychia* spp. (27%), *U. lactuca* (12%)*, U. intestinalis* (10%) and *Porphyra* spp. (2%) were the variables that best explained the presence of *T. magellanicus* (adjusted R^2^ = 0.52).

## 4. Discussion

Our survey data confirm that *T. magellanicus* is distributed across the northern coast of Navarino Island. Although every bay along the coast could not be examined, our data suggest that the species is present throughout this area, while recent opportunistic observations in the Argentinian Tierra del Fuego and the Falkland Islands support an as yet undocumented wider regional distribution. Across the surveyed bays, it was clear that the favoured microhabitats for the larvae were locations where the algae *Bostrychia* spp. and *Ulva* spp. were present, but also where there was a combination of these algae with *Porphyra* spp., boulders, gravel, sand and bivalves. It is very likely that algae, especially the more filamentous taxa such as *Bostrychia* spp., desiccate more slowly than the exposed substrata, maintaining higher R.H. levels during the low-tide period. In some of the bays, instead of an abundant presence of *Bostrychia* spp., there was greater presence of *U. intestinalis* or *U. prolifera*, which were also good predictors for the presence of *T. magellanicus*.

One of the most striking features of the adults was their extremely active behaviour, which suggests high energy expenditure in a very short period of time. This is likely driven by the limited amount of time they have to find mates in the period between successive high tides. This is similar to reports of the behaviour of *Pontomyia* spp., another intertidal chironomid genus found in the Caribbean, north-eastern Brazil, Japan, south-east Asia and Australia [47,48,49,58,59], and which has one of the shortest known insect adult lifespans (maximum of three hours for the winged males and the vermiform females) [54]. Even though adults were seen mating in the field, we did not directly observe oviposition but, given the distribution of larvae, this is likely to take place on or within algal mats.

The absence of both adults and larvae in the high intertidal zone is likely associated with the lack of suitable microhabitat to escape from extremes of microclimate in this area. Across all bays, this zone primarily consists of boulders and sand mixed with clay, making the substrate too hard for the larvae to burrow into, and thereby exposing them to dangerous stressors such as sunlight/UV exposure and wind, as well as potential predators (at least one species of insectivorous bird, *Lessonia rufa* (Gmelin, 1789), frequently forages in the intertidal zone of Róbalo Bay, and other bird species, including *Vanellus chilensis* (Molina, 1782) and *Xolmis pyrope* (Kittlitz, 1830), are also often present in the area). It was unclear which variables directly restricted the presence of the species in the low-tide zone, but this zone is covered by seawater for most of the time, being water-free for only 1–2 h during the low tide. This zone may, therefore, be hostile to the adults due to the risk of drowning, a hypothesis that is consistent with our casual observations of the high mortality of adults when an area is inundated. A caveat for some of these hypotheses is that our environmental measurements of temperature, R.H. and wind speed were taken at different times of the day and across different days. However, these differences are likely to add noise to the data rather than leading to a systematic difference in readings being recorded between the different zones.

The mid-tide zone contained by far the highest numbers and density of *T. magellanicus*, whose larvae were abundant in filamentous algae growing on different substrates, such as bivalves and boulders. Within the mid zone, the variables that were most consistently associated with the presence of *T. magellanicus* were boulders, stones, *Bostrychia* spp. and *U. intestinalis*, consistent with the findings of the more general distribution study (Table 3). It is likely that these environmental features provide good shelter from extremes of temperature, irradiation and wind exposure, either by direct physical protection from these factors or by creating microclimates within, thereby reducing desiccation stress.

## 5. Conclusions

The intertidal zone is the key habitat type for *T. magellanicus*, but within this zone certain conditions support higher larval population densities, namely large rocks (i.e., boulders) and filamentous algae (particularly *Bostrychia* spp. and *Ulva intestinalis*). As a result, the species’ overall distribution is widespread but patchy. The data presented lay the foundation for future studies of this unusual insect’s distribution, ecology, phenology, physiology and general biology. With improved knowledge of the species’ preferred microhabitats, we are better able to predict other locations where it may occur, and even to extrapolate the risk of it being transported elsewhere (e.g., in species distribution modelling, which can involve the direct use of GIS data or a combination of biogeographical, physiological and meteorological data [33,60,61,62]).

## Figures and Tables

**Figure 1 insects-11-00442-f001:**
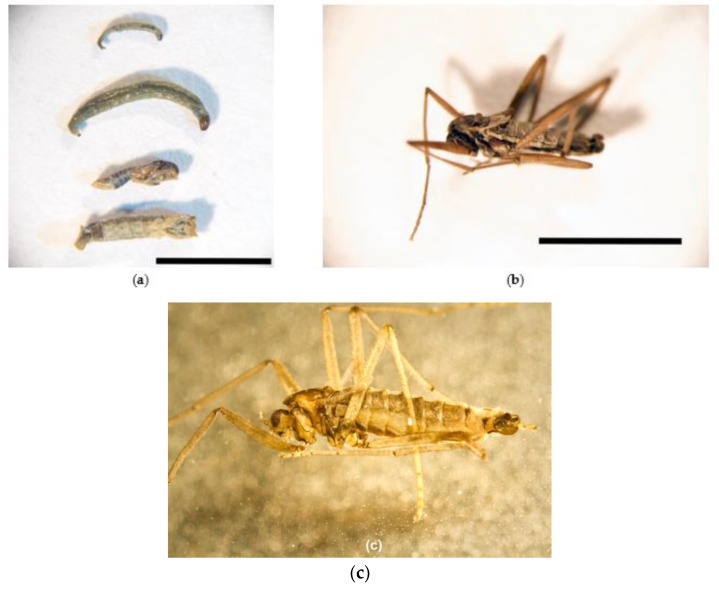
Specimens of *Telmatogeton magellanicus*: (**a**) Two larvae (2nd and 4th instars) and two pupae (one larger female and one smaller male); (**b**,**c**) Adult females ((**c**) by Gonzalo Arriagada). Scale bar = 0.5 cm.

**Figure 2 insects-11-00442-f002:**
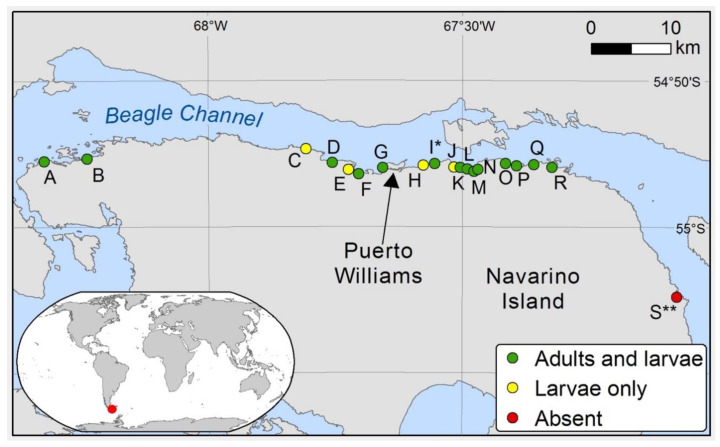
Navarino Island with the bays surveyed, colour-coded for presence/absence of adults and larvae of *Telmatogeton magellanicus*. (A, Puerto Navarino; B, Honda Bay; C, “Bahia Linda”; D, Chicha de Pera; E, “Second” Bay; F, Los Bronces; G, Róbalo Bay; H, Ukika; I, Puente Guanaco Bay; J, Punta Truco; K, “Seventh” Bay; L, Amarilla Bay; M, “Fifth” Bay; N, Corrales Bay; O, “King Penguin” Bay; P, “Third” Bay; Q, “Casita” Bay; R, Eugenia Bay; S, Puerto Toro) (* also *Telmatogeton* spp.; ** survey time was very limited).

**Figure 3 insects-11-00442-f003:**
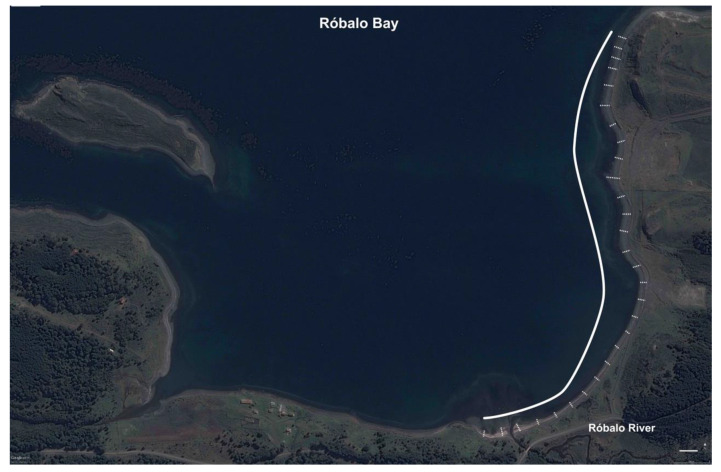
The location of the habitat transects at Róbalo Bay. Each transect included a quadrat (60 cm × 60 cm) deployed in the high-tide, mid-tide and low-tide zones of the shoreline. Satellite image edited from Google Earth (earth.google.com/web/).

**Figure 4 insects-11-00442-f004:**
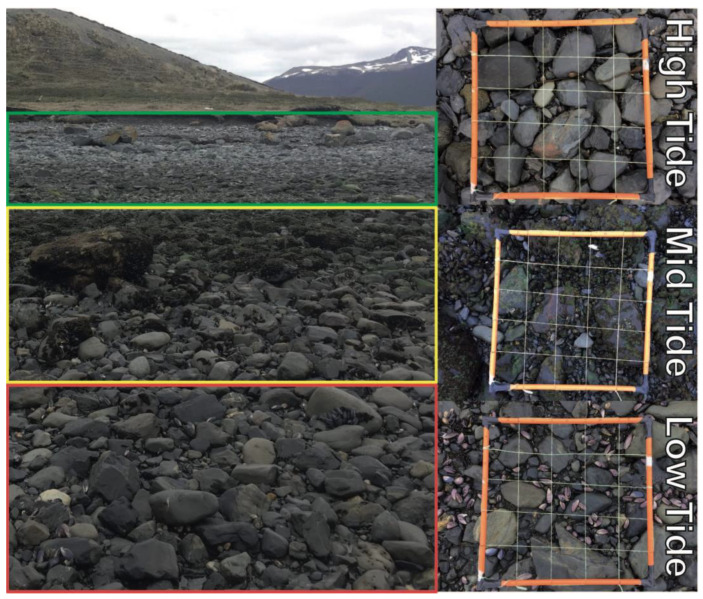
The three tidal areas in Róbalo Bay, with the representative quadrats (60 cm × 60 cm) used for habitat characterisation.

**Figure 5 insects-11-00442-f005:**
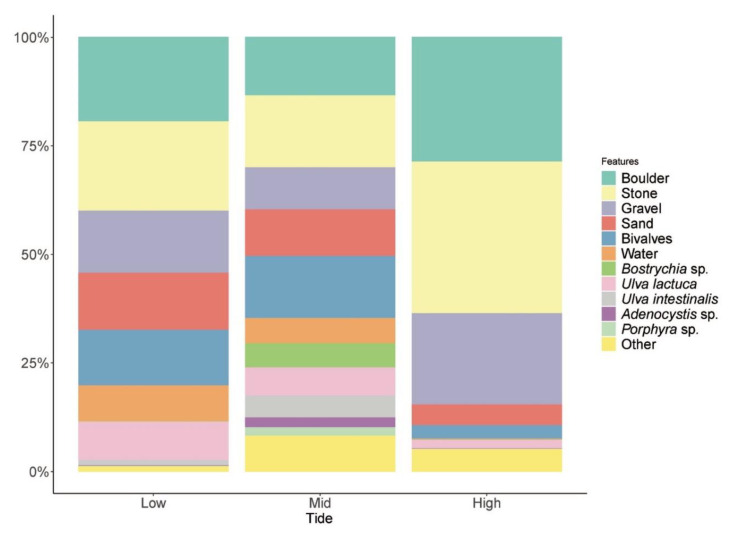
Stacked histogram with the distribution of the 12 measured environmental variables across the three tidal zones (low, mid and high).

**Figure 6 insects-11-00442-f006:**
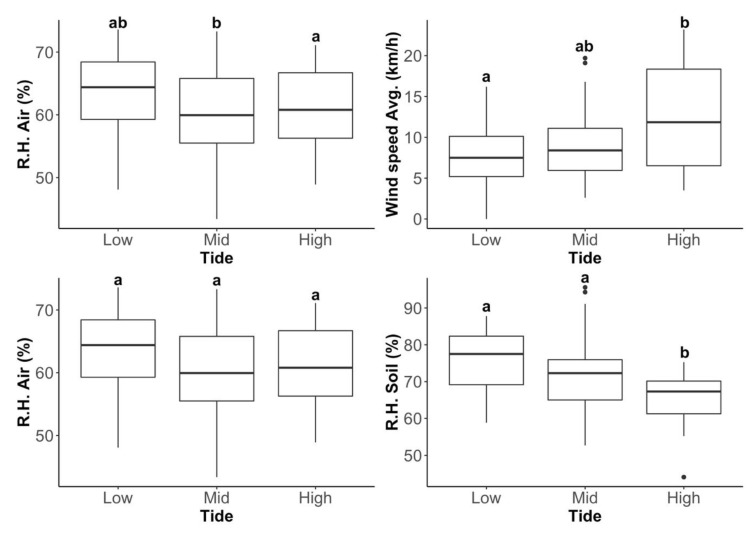
Air temperature, average wind speed, as well as air and substrate R.H. across the three tidal zones (low, mid and high). Means with the same letter are not significantly different at *p* < 0.05 (Pairwise Wilcoxon Rank Sum Tests).

**Figure 7 insects-11-00442-f007:**
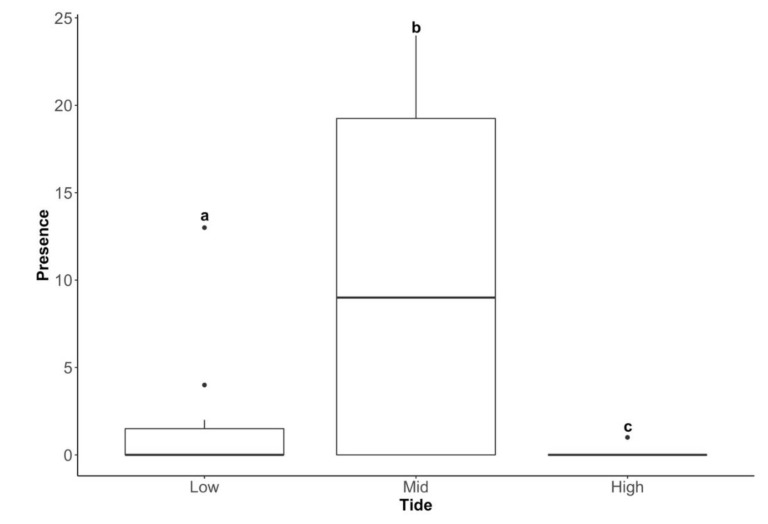
Presence of larvae of *T. magellanicus* in quadrats (25 sub-quadrats per quadrat) across the three tidal zones (low, mid and high). Means with the same letter are not significantly different at *p* < 0.05 (Pairwise Wilcoxon Rank Sum Tests).

**Table 1 insects-11-00442-t001:** PERMANOVA post hoc tests for environmental conditions between the different tidal zones in Róbalo Bay, Navarino, Chile. Asterisks indicate significant differences (** *p* < 0.01, *** *p* < 0.001).

Pairwise Test	Environmental Conditions (12 Variables)		
	t	Unique Perms	*p* (Perm)
High vs. Low	2.7567	9939	0.0001 ***
High vs. Mid	2.8884	9925	0.0001 ***
Low vs. Mid	1.723	9934	0.0011 **

**Table 2 insects-11-00442-t002:** Similarity Percentages (SIMPER) analysis of the environmental variables that contributed 90% of the habitat composition within the three tidal zones studied in Róbalo Bay, Navarino Island, Chile (Contrib.% = percentage of variable contribution; Cum.% = cumulative contribution).

Tide Level	Environ. Variable	Av. Value	Sq. Dist/SD	Contrib.%	Cum.%
Low	*Bostrychia* spp.	0.04	0.20	0.01	0.01
	*Adenocystis* spp.	0.04	0.20	0.01	0.01
	*Porphyra* spp.	0.04	0.20	0.01	0.02
	Other	1.00	0.37	0.54	0.56
	*Ulva intestinalis*	0.93	0.26	1.19	1.75
	Water	6.26	0.50	7.12	8.87
	Gravel	10.80	0.57	13.95	22.82
	*Ulva Lactuca*	6.63	0.49	14.22	37.04
	Stone	15.6	0.53	14.36	51.40
	Boulder	14.7	0.54	14.70	66.11
	Bivalves	9.7	0.55	15.62	81.73
Mid	*Porphyra* spp.	1.75	0.31	2.97	2.97
	*Adenocystis* spp.	2.18	0.28	4.45	7.41
	Water	5.43	0.44	5.41	12.82
	*Bostrychia* spp.	5.25	0.43	6.48	19.30
	*Ulva Lactuca*	6.11	0.45	6.63	25.93
	*Ulva intestinalis*	4.68	0.42	7.67	33.60
	Stone	15.60	0.51	7.77	41.38
	Boulder	12.60	0.54	8.08	49.46
	Gravel	9.18	0.56	10.67	60.12
	Sand	10.10	0.55	12.42	72.54
	Bivalves	13.5	0.57	13.32	85.86
High	*Porphyra* spp.	0.036	0.19	0.01	0.01
	*Bostrychia* spp.	0.71	0.27	0.02	0.04
	*Adenocystis* spp.	0.71	0.19	0.05	0.09
	Water	0.11	0.19	0.12	0.21
	*Ulva Lactuca*	1.18	0.31	3.10	3.31
	Stone	22.20	0.45	4.20	7.51
	Bivalves	2.00	0.26	8.93	16.43
	Sand	3.04	0.40	13.54	29.97
	Other	3.32	0.34	16.60	46.57
	Gravel	13.4	0.54	24.84	71.41

**Table 3 insects-11-00442-t003:** Marginal tests obtained from the Distance-Based Linear Modelling procedure for the 12 environmental variables measured within the three tidal zones in Róbalo Bay, Navarino Island, Chile. Asterisks indicate significant associations with the presence of *T. magellanicus* (* *p* < 0.05, ** *p* < 0.01, *** *p* < 0.001), and values in bold the most highly associated variables. (Prop. = proportion of variability).

Variable	Pseudo-F	*p*	Prop.
Boulder	4.39	0.03 *	0.05
Stone	0.02	0.91	0.00
Gravel	1.78	0.16	0.02
Sand	8.17	0.003 **	0.09
Bivalves	11.42	0.002 **	**0.12**
Water	1.52	0.24	0.02
*Bostrychia* spp.	30.12	0.001 ***	**0.27**
*Ulva lactuca*	10.68	0.004 **	**0.12**
*Ulva intestinalis*	9.48	0.01 *	**0.10**
*Adenocystis* spp.	4.77	0.04 *	0.06
*Porphyra* spp.	1.74	0.13	0.02
Other	0.03	0.86	0.00

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
