# Peer review of "Distribution and Habitat Preferences of the Newly Rediscovered Telmatogeton magellanicus (Jacobs, 1900) (Diptera: Chironomidae) on Navarino Island, Chile"

_insects, 2020, doi:10.3390/insects11070442_

Round 1

Reviewer 1 Report

The manuscript “Distribution and habitat preferences of the newly rediscovered Telmatogeton magellanica (Jacobs, 1900) (Diptera: Chironomidae) on Navarino Island, Chile” quantitatively describes the habitat of an unusual midge species. This intertidal species and its habitat is an interesting study subject and the determination of part of its autecology is important to understanding the role of such species within the regional environment and how they can be conserved. The revisions made by the authors are appropriate and improve the manuscript. I recommend that this manuscript be accepted for publication in Insects in its present form.

Author Response

Dear Reviewer 1,

Thank you very much for reviewing our manuscript.

All adjustments have been made

Kind regards

Reviewer 2 Report

I am satisfied with the new version of the manuscript and I suggest to publish it in the present form.

Author Response

Dear Reviewer 2,

Thank you very much for reviewing our manuscript.

All adjustments have been made

Kind regards

Reviewer 3 Report

This updated draft is ready to publish with the exception of the following:

Line 123, 272. Not sure what “opportunistic” notes means. Perhaps you mean that these are simply observations rather than data you analyzed? Please clarify.

Author Response

Dear Reviewer 3,

Thank you very much for reviewing our manuscript.

All minor adjustments have been made, including:

Line 123, 272. Sentence changed to "We also made notes of any aspects of adult behaviour that we observed on occasion."

Kind regards!